# Synthesis of Magnetic Ferrocene-Containing Polymer with Photothermal Effects for Rapid Degradation of Methylene Blue

**DOI:** 10.3390/polym13040558

**Published:** 2021-02-13

**Authors:** Wenhui Zhu, Caiyun Zhang, Yali Chen, Qiliang Deng

**Affiliations:** College of Chemical Engineering and Materials Science, Tianjin University of Science and Technology, Tianjin 300457, China; 15735151231@163.com (W.Z.); zcyyhdjc@163.com (C.Z.); chenyaliais@foxmail.com (Y.C.)

**Keywords:** ferrocene-containing polymer, photothermal, Fenton reaction, methylene blue

## Abstract

Photothermal materials are attracting more and more attention. In this research, we synthesized a ferrocene-containing polymer with magnetism and photothermal properties. The resulting polymer was characterized by Fourier-transform infrared (FT-IR), vibrating sample magnetometer (VSM), scanning electron microscope (SEM), energy dispersive X-ray spectroscopy (EDS), X-ray photoelectron spectroscopy (XPS), X-ray diffraction (XRD), and thermogravimetric analysis (TGA). Its photo-thermocatalytic activity was investigated by choosing methylene blue (MB) as a model compound. The degradation percent of MB under an irradiated 808 nm laser reaches 99.5% within 15 min, and the degradation rate is 0.5517 min^−1^, which is 145 times more than that of room temperature degradation. Under irradiation with simulated sunlight, the degradation rate is 0.0092 min^−1^, which is approximately 2.5 times more than that of room temperature degradation. The present study may open up a feasible route to degrade organic pollutants.

## 1. Introduction

Organic photothermal conversion materials have attracted intensive attention due to their prominent property that can transform infrared light into heat, and have displayed great potential applications in many fields, such as photothermal/photoacoustic imaging, photothermal therapy, photothermal killing of bacteria, photothermal-electric devices, and shape-memory devices [1]. In addition, some photothermal materials have been used for water treatment, which showed a positive effect on repairing the environment. However, few reports have combined these photothermal properties with the Fenton reaction to degrade organic pollutants. 

Besides adsorption separation [2,3,4,5], the Fenton reaction has been extensively explored in the field of the removal of hazardous organic pollutants in water, which can directly degrade the contaminant into harmless inorganic salts, water, or carbon dioxide [6]. Most of this research about the photo-Fenton reaction for organic pollutants is mainly focused on UV and visible light; however, 54% of the solar spectrum is near-infrared (NIR) light, which is rarely utilized [7]. Generally, common materials with photothermal effects include noble metals, organic compounds, carbon-based materials, aluminum nanoparticles, metallic oxide, and metal sulfide [8]. The photothermal conversion performance of materials can be improved by covalently linking the electron-donor, extending the molecular conjugation length and electron-acceptor fragments, and inhibiting the radiative transition process [6].

Ferrocene is regarded as having a sandwich-like molecular configuration and an organometallic compound that is highly inert owing to the great dissociation energy iron-cyclopentadienyl (91 cal/mol) [9], which has been employed in a wide range of areas such as electronic, electrochemical, nanomedicine, and biological sensing, and as a catalyst. Ferrocene-containing polymers have characteristic features, such as excellent redox activity, high chemical, and thermal stability, and are becoming an important kind of material. Different types of ferrocene-containing polymers have been synthesized by various polymerization strategies and post-modification methods [10]. Among them, ferrocene-containing polymers with an alternating conjugated aromatic segment are usually synthetically challenging [11], and they has been prepared through polycondensation, intermolecular coupling, and cross-metathesis. In these polymerization methods, the polycondensation approach is a fast and facile synthesis method involving the interaction of two defunction grounds or same-type molecules [10,12].

Herein, a new ferrocene-containing polymer was synthesized via a polycondensation approach. 1,1′-Ferrocenedicarboxaldehyde and 4,4′-diamino-p-terphenyl were connected via a facile imine condensation reaction to afford the desired conjugated polymer. To investigate the catalytic and photothermal performance of the resulted polymer, the adsorption and degradation of methylene blue (MB) was studied.

## 2. Experimental Section

### 2.1. Reagents and Chemicals

1,1′-Ferrocenedicarboxaldehyde (97.0%), dioxane (99.0%), and mesitylene (98.0%) were purchased from Aladdin Reagent Co. Ltd. (Shanghai, China). 4,4′-diamino-p-terphenyl and MB were obtained from J&K Chemical (Beijing, China). Methanol was purchased from Fuchen Chemical Reagent Co. Ltd. (Tianjin, China). *N*,*N*-Dimethylformamide (DMF) and hydrogen peroxide (H_2_O_2_, 30.0 wt%) were purchased from Damao Chemical Reagent Factory (Tianjin, China). Acetic acid (CH_3_COOH, 36.0%) was purchased from the Institute of Guangfu Fine Chemical (Tianjin, China). All the available reagents and chemicals were used without further purification.

### 2.2. Instruments

Scanning electron microscopy (SEM) images were obtained by a field emission scanning electron microscope (Quanta FEG 250). The corresponding energy-dispersive X-ray spectroscopy (EDS) was carried out on an Oxford INCA X-MAX50 (Oxford, UK). X-ray photoelectron spectroscopy (XPS) experiments were carried out on an Escalab 250xi photoelectron spectrometer (Thermo, Boston, Massachusetts, USA). The magnetization curve was determined by a commercial magnetic property measurement system (Squid-VSM, Quantum, San Diego, California, USA). Fourier-transform infrared (FT-IR) spectra (4000 cm^−1^–400 cm^−1^) in KBr were acquired on a Vector 22 FT-IR spectrophotometer (Bruker, Germany). A thermogravimetric analysis (TGA) was performed using Universal V4.5A TA Instruments (TA, Newcastle, Delaware, USA) with a ramp of 10 K min^−1^ in nitrogen atmosphere between 40−800 °C. The crystalline properties of polymer were determined by X-ray diffraction (XRD, D8 Advance Bruker-AXS, Bruck, Germany) analysis using Cu–Kα radiation. The concentrations of MB solution were detected by a UV-visible spectrophotometer (UV-Vis, TΜ-1901, Puxi, China). Other instruments were used including a simulated sunlight xenon lamp light source system (Beijing Bofeilai Technology Co., Ltd., Beijing, China, PLS-SXE300D) and an optical-fiber-coupled 808 nm diode-laser (Changchun New Industries photoelectric technology Co. Ltd., Changchun, China).

### 2.3. Fabrication of Polymer

1,1′-Ferrocenedicarboxaldehyde (24.2 mg, 0.10 mmol), 4,4′-diamino-p-triphenyl (31.2 mg, 0.12 mmol), dioxane (0.5 mL) and mesitylene (0.5 mL) were added into a Pyrex tube (10 mL). After the mixture was sonicated for 5 min, acetic acid (0.2 mL, 1 M) was added. The system was sonicated for 15 min. The reaction tube was sealed with a triple freezing–thawing cycle, and the reaction was maintained at 180 °C for 72 h. The product was collected by filtrating, and washing with DMF and CH_3_OH until the supernatant became clear. Then the brownish–black powder was obtained after being drying at 100 °C. 

### 2.4. Adsorption Performance of the Materials to MB

In order to evaluate the adsorption property of polymer, all adsorption experiments were carried out under dark conditions. For isothermal adsorption and adsorption kinetics experiments, 2.0 mg of magnetic polymer was added to 2.0 mL of the known concentration of MB dye solution. In the adsorption isothermal experiments, the concentrations of MB ranged from 40.0 to 500.0 mg L^−1^ and the incubation time was 24 h. In adsorption kinetics experiments, 400.0 mg L^−1^ of MB solution was selected. The absorption of MB solution was checked at 664 nm. The adsorption capacities of the resulting polymer were obtained according to the following equation [13,14]:(1)$$q_{e}=\frac{\left(C_{0}{-C}_{e}\right)V}{m}$$

The adsorption amount (q_t_) was calculated by
(2)$$q_{t}=\frac{\left(C_{0}{-C}_{t}\right)V}{m}$$
where q_e_ and q_t_ represent the equilibrium adsorption capacity and the adsorption amount at time t, respectively, C_0_, C_e,_ and C_t_ are the MB concentration at initial time, equilibrium and time t, respectively, V is the volume of the solution, and m is the mass of the adsorbent.

### 2.5. Photo-Thermocatalytic Degradation Experiment

After adsorption equilibrium, the polymer was added to 1000.0 mg L^−1^ of MB solution. The degradation experiment was performed in the presence of H_2_O_2_ under the irradiation of an NIR (808 nm, 2 w cm^−2^) or 300 W Xe lamp. The catalytic activity was confirmed by measuring the changes in the absorption at different time intervals using a UV–vis spectroscopy at 635 nm. The experiment was also performed at 25 °C under dark conditions. 

## 3. Results and Discussion

### 3.1. Preparation and Characterization

The polymer was synthesized by the condensation reaction between 1,1′-ferrocenedicarboxaldehyde and 4,4′-diamino-p-triphenyl via the formation of imine linkages (Scheme 1). The morphological structures of the polymer were characterized via SEM (Figure 1). The SEM image showed flocculent structure. The EDS images showed the uniform distributions of the C, N, Fe, and O elements without supplementary impurities in the polymer (Figure 2). The chemical structure of polymer was characterized by FT-IR (Figure 3a). The FT-IR spectrum of 1,1′-Ferrocenedicarboxaldehyde showed a significant peak at 1675 cm^−1^ for HC=O stretching vibration. The FT-IR spectrum of 4,4′-diamino-p-terphenyl showed NH_2_ stretching and bending vibration at 3443 cm^−1^, and 1606 cm^−1^, respectively. The polymer showed characteristic stretch bands at 460–578, 740–878, 1485, and 2974 cm^−1^, which were assigned to Cp-Fe, C–H bending modes, C=C stretch, and C-H stretching vibration in Cp, respectively. The typical peaks for C=N vibration of polymer at 1610 cm^−1^ indicated imine bond formation in the polymer. The peaks at 1675 cm^−1^ and 3372 cm^−1^ were assigned to the remaining terminal aldehydes and N-H stretch band, respectively. The magnetic properties of the resulting polymer were measured in the field range from −20,000 to 20,000 Oe (Figure 3b). The saturation magnetization of the polymer was approximately 8.7 emu g^−1^, which indicated that the magnetic property of polymer could be used for magnetic separation. The reason may be due to the expected magnetic coupling between the alternate donor and recipient under strict superposition conditions. From the TG curve (Figure 3c), we could observe that the solvent in the polymer was evaporated at approximately 100 °C. A gently thermolytic degradation in the temperature range of 100–660 °C was observed. Subsequently, a rapid thermal degradation occurred. The mass reduction at 800 °C was about 76.04%. The DSC curve shows the main steps for the decomposition of polymer and the maximum rate of degradation were at 660°C, which indicates excellent thermal stability. In Figure 3d, a single low intensity broad peak appeared at 2θ = 18.5°, which is characteristic of the amorphous nature of the resulting polymer [15]. This is similar to the peak location of semi-crystalline ferrocenyl polymers [16]. The chemical structure components of the sample surfaces were determined by XPS. The intensive lines of C 1s, N 1s, O 1s, and Fe 2p corresponded to the photoelectron peaks of each element (Figure 4.). The C 1s peak in the spectra of the resulting polymer shows the four components at the binding energies of 284.6, 285.1, 286.3, 287.3, and 290.4 eV, which was due to the functional groups of C-C/C-H, C=C, C-N, C=N, and C=O [17,18,19,20,21], respectively. In addition, the peaks at 398.6, 399.7, and 402.8 eV were related to the chemical environments of C=N, C-N, and N-H [22,23,24], respectively. The peak of O 1s centered at 532.1 eV corresponded to the C=O group [25]. The Fe 2p XPS signals located at 707.7, and 720.5 eV belonged to Fe(Ⅱ) 2p_3/2_ and Fe(Ⅱ) 2p_1/2_ in the ferrocene unit [25,26,27]. The peaks at 711.6 and 726.7 eV corresponded to Fe(Ⅲ) 2p_3/2_, and Fe(Ⅲ) 2p_1/2_ [28], respectively.

### 3.2. Photothermal Property

The photothermal effect of the resulting polymer was measured by dispersing it in water under irradiation with a near infrared 808 nm laser with a power density of 2 W m^−2^. The temperature of the resulting polymer increased rapidly, recorded by an IR thermal camera. As shown in Figure 5a, the temperature of the resulting polymer sharply increased and reached 51.1 °C within 600 s. On the other hand, the temperature of the water in the absence of the polymer improved only 3.6 °C. The cooling curve was also monitored (Appendix A), and the calculated conversion efficiency was 19.25% (details are shown in the Appendix A). The temperature response of the polymer suspension measured over five repeated irradiation cycles showed a perfect cyclability of maximum temperature (Figure 5b). These results confirmed the excellent photothermal conversion efficiency. The reason might be attributed the Fc, which as an excellent electron-donating unit can inhibit singlet-oxygen (^1^O_2_) production and quench fluorescence emission by a photo-induced electron transfer (PET) process, and thus improve the nonradiative transition of thermal energy release [29]. Furthermore, it has also been proven that iron-containing polymer is considered as a new class of photothermal agents [30]. 

### 3.3. Adsorption Property

The resulted polymer as an adsorbent for MB was evaluated by measuring adsorption isotherms and adsorption kinetics. As shown in Figure 6a, the adsorption capacity of the resulting polymer increased with the variation of MB concentration from 40.0 to 400.0 mg L^−1^. When the concentration exceeded 400 mg L^−1^, the adsorption capacity maintained constant, at 42.28 mg g^−1^. The results could be explained by the N atom and MB coordination to the Fe element and strong π-π interactions. The adsorption isotherms can be used to describe the types of interactions between the targets and adsorbents. Thus, the adsorption isotherms mechanism of MB on the resulting polymer were analyzed to fit the experimental data with the Langmuir, Freundlich, and Temkin isotherms models, respectively. The corresponding linear equations are presented below (Equations (3)–(5)) [31],
(3)$$\frac{C_{e}}{q_{e}}=\frac{C_{e}}{q_{m}}+\frac{1}{K_{L}q_{m}}$$
(4)$${\mathrm{lnq}}_{e}{=\mathrm{lnK}}_{F}+\frac{1}{n}{\mathrm{lnC}}_{e}$$
(5)$$q_{e}{=B}_{T}{\mathrm{lnK}}_{T}{+B}_{T}{\mathrm{lnC}}_{e}$$
where q_e_ (mg g^−1^) is adsorption capacity and C_e_ (mg L^−1^) is concentration at equilibrium. q_m_ (mg g^−1^) and K_L_ (L g^−1^) are the maximum adsorption capacity and the Langmuir constant, respectively. K_F_ (L mg^−1^) and n represents Freundlich adsorption empirical constants. B_T_ = RT/b_T_, and b_T_ and R are constants related to the energy of adsorption and gas constants, respectively. K_T_ is the Temkin equilibrium constant.

The fitting results arre given in Figure 6b and Appendix A. According to the correlation coefficients, the experiment results matched best with the Langmuir model best, which indicated the monolayer coverage of the MB onto the adsorbent surface.

In addition, the adsorption rate and kinetic mechanism of MB onto the resulting polymer are also highly important properties (Figure 6c). In order to investigate the adsorption kinetics, several models (pseudo-first order, pseudo-second order, intra-particle diffusion, and elovich) were adopted. The corresponding simulations were finished by the following Equations (6)–(9) [32], respectively,
(6)$$q_{t}{=q}_{e}{(1-e}^{{-k}_{1}t})$$
(7)$$q_{t}=\frac{q_{e}^{2}k_{2}t}{{1+q}_{e}k_{t}}$$
(8)$$q_{t}=\frac{1}{\beta }\ln\left(\alpha \beta \right)+\frac{1}{\beta }\ln$$
(9)$$q_{t}{=k}_{3}t^{\frac{1}{2}}+C$$
where q_t_ and q_e_ (mg g^−1^) represent adsorption capacity at time t (min) and equilibrium time. k_1_ (min^−1^) is the rate constant of the pseudo-first order, k_2_ (g mg^−1^ min^−1^) and k_3_ (mg g^−1^ min^−1/2^) represent the pseudo-−second order and intra-particle diffusion, respectively. α (mg g^−1^ min^−1^) and β (g mg^−1^) represent the initial absorption rate and a desorption constant of the Elovich model, respectively. C (mg g^−1^) represents a constant related to the intraparticle diffusion model. 

As shown in Figure 6d and Appendix A, the pseudo-first order model was best suitable for the adsorption kinetics process of the polymer to MB in terms of the correlation coefficient, which suggests that the physical adsorption dominates the adsorption process.

### 3.4. Catalytic Property

Since H_2_O_2_ is easily activated to generate hydroxyl radicals in the presence of iron, the Fe^2+^/Fe^3+^ redox cycle is carried out on the surface of the catalyst. Thus, the Fenton reaction has an obviously catalytic degradation effect for organic pollutants, which results from high oxidation potential due to the formation of hydroxyl [5,33]. The ratio of the MB concentration (C_t_) at time t (min) to the initial concentration (C_0_) was used to investigate the catalytic performance upon different degradation time (Figure 7a). The degradation of MB in the presence of the polymer can reach 98.4% within 21 h under room temperature, and the k is 0.0038 min^−1^. Although an efficient catalytic degradation property was confirmed at room temperature, based on its excellent photothermal effectiveness, an NIR laser was used to irradiate the system. A series of MB concentrations was selected for the investigation of degradation within 30 min. The removal efficiencies of MB reached 98.7–99.9% (Figure 7b). In order to further investigate the catalytic property, the MB concentration at 1000 mg L^−1^ as a target to study the degradation over time. The change of the MB concentration during the photo-thermo catalytic is nonlinear and the degradation rate slows gradually. Therefore, the fastest degradation stage was selected to investigate within 5–10 min, and k reached 0.5517 min^−1^. The corresponding curves of the degradation rate and temperature indicated the degradation could finish within 15 min; the temperature rose to the maximum, and then dropped to a plateau (Figure 7c). In addition, the catalytic activities of the resulting polymer for the degradation of MB were also evaluated under the simulated sunlight with a power density of 1 kW m^−2^. When MB was illuminated with a simulated solar illumination for 150 min, the degradation of MB (C_t_/C_0_) reached 0.313, and k was 0.0092 min^−1^ (Figure 7d). The temperature of the solution enhanced from 28 °C to 35 °C, which was attributed to the effective light-to-thermal conversion performance and excellent solar absorption (Appendix A). In order to accelerate the degradation of MB, the sample was illuminated with light of 3 kW m^−2^ for 25 min, C_t_/C_0_ reached 0.0195, and k was 0.1887 min^−1^ (Figure 7e). According to the recorded temperature, the surface temperature improved to 55°C (Appendix A). 

The reusability of polymer was examined by repeating the Fenton reaction for five cycles under the same operating conditions. The degradation of the MB for the five consecutive cycles with degradation percentage values reached approximately 81.64–90.25% within 10 min; RSD was 4.15% (Figure 7f). These results confirm the good reusability of the resulting polymer.

## 4. Conclusions

In summary, a ferrocene-based organic magnetic polymer with photothermal properties was synthesized. NIR and simulation sunlight irradiation approaches in promoting the degradation of MB in the Fenton reaction were investigated. The efficient photo-thermocatalytic activity in the degradation of MB under 808 nm laser irradiation indicated that MB could be completely removed within 30 min. The resulting polymer also displayed excellent photo-degradation efficiency (C_t_/C_0_ = 0.313, and k = 0.0092) under the simulated sunlight as an extra energy source. The results confirmed that a relatively high degradation effect took place in the presence of the resulting polymer. Finally, we believe that adding the magnetic organic polymer in the Fenton reaction is a promising strategy for clear water production. 

## Data Availability

The data presented in this study are available on request from the corresponding author.

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
