# Peer review of "Synthesis of Magnetic Ferrocene-Containing Polymer with Photothermal Effects for Rapid Degradation of Methylene Blue"

_polymers, 2021, doi:10.3390/polym13040558_

Round 1
Reviewer 1 Report
The manuscript by Deng and co-workers describes the synthesis and application of ferrocene-base polymers. Dye removal from water using polymers is an important and timely topic and fits the scope of the journal. The authors presented a good amount of the data for the polymer and the application but there are still numerous minor and major issues before the manuscript can be further considered for publication in Polymers.
1, The rationale for the selection of the concentration range should be given in the manuscript. The concentration range seems orders of magnitude higher than what is practically relevant. Justification should be given, and a short discussion added.
2, The clean-up of organic contaminants in water is a timely and important topic, and emerging approaches using polymers should be briefly acknowledged (DOIs 10.1016/j.apmt.2020.100878; 10.1021/acssuschemeng.0c00129; 10.1039/D0GC01709D; 10.1021/acssuschemeng.9b07026).
3, Have the authors considered characterizing the degradation products? Are they less harmful in the environment than the dye itself?
4, The reusability of the materials should demonstrated of 5-10 cycles with respect to both the performance of the polymers as well as their structural integrity. The performance is already presented in Figure 7f but it is unclear how the reported errors were derived.
5, The polymers should be better characterized. The molecular weight of the synthesized polymers should be reported. What is the atomic percentage of the iron in the polymer, and how does it compare to the theoretical structure presented in Scheme 1?
6, The conclusion section should summarize the main research findings in quantitative statements as well.
7, The reference list has multiple typos, missing page numbers etc, proofread the list.
Author Response
1, The rationale for the selection of the concentration range should be given in the manuscript. The concentration range seems orders of magnitude higher than what is practically relevant. Justification should be given, and a short discussion added.
Answer: Thanks for your suggestion. The higher concentration was chosen in order to evaluate the catalytic property. The discussion has been added to the revised manuscript.
2, The clean-up of organic contaminants in water is a timely and important topic, and emerging approaches using polymers should be briefly acknowledged (DOIs 10.1016/j.apmt.2020.100878; 10.1021/acssuschemeng.0c00129; 10.1039/D0GC01709D; 10.1021/acssuschemeng.9b07026).
Answer: Thanks for your suggestion. These references have been cited in the revised manuscript.
3, Have the authors considered characterizing the degradation products? Are they less harmful in the environment than the dye itself?
Answer: Thanks for your suggestion. Indeed, the characterizations of the degradation products are important, however, we didn’t perform the experiments. Now, it is very difficult to perform the experiments, because we are in the spring-festival holiday, and our labs have been closed.
4, The reusability of the materials should demonstrated of 5-10 cycles with respect to both the performance of the polymers as well as their structural integrity. The performance is already presented in Figure 7f but it is unclear how the reported errors were derived.
Answer: Thanks for your suggestion. The reusability of the materials has been investigated in the revised manuscript (Figure 7f). The reported errors produced from the triplicate experiments.
5, The polymers should be better characterized. The molecular weight of the synthesized polymers should be reported. What is the atomic percentage of the iron in the polymer, and how does it compare to the theoretical structure presented in Scheme 1?
Answer: Thanks for your suggestion. Now, it is very difficult to perform the experiments, because we are in the spring-festival holiday, and our labs have been closed.
6, The conclusion section should summarize the main research findings in quantitative statements as well.
Answer: Thanks for your suggestion. The conclusion section has been improved in the revised manuscript.
7, The reference list has multiple typos, missing page numbers etc, proofread the list.
Answer: Thanks for your suggestion. The reference list has been corrected in the revised manuscript.

Reviewer 2 Report
Well organized and well presented work. The authors studied and characterized their material with various technics. The manuscript could published after these revisions.
- Arrange the equations 1-7.
- What about statistical analysis of the data; A statistical analysis should provided in the revised version.
Author Response
- Arrange the equations 1-7.
Answer: Thanks for your suggestion. The equations 1-7 have been arranged in the revised manuscript.
- What about statistical analysis of the data; A statistical analysis should provided in the revised version.
Answer: Thanks for your suggestion. The statistical analysis has been provided in the revised manuscript.
